# Strategies to Enhance CO_2_ Electrochemical Reduction from Reactive Carbon Solutions

**DOI:** 10.3390/molecules28041951

**Published:** 2023-02-18

**Authors:** Carlos Larrea, Juan Ramón Avilés-Moreno, Pilar Ocón

**Affiliations:** Departamento de Química Física Aplicada, Universidad Autónoma de Madrid (UAM), C/Francisco Tomás y Valiente 7, 28049 Madrid, Spain

**Keywords:** bicarbonate electrochemical reduction, CO_2_ electrolysis, carbon capture and utilization, silver catalyst

## Abstract

CO_2_ electrochemical reduction (CO_2_ ER) from (bi)carbonate feed presents an opportunity to efficiently couple this process to alkaline-based carbon capture systems. Likewise, while this method of reducing CO_2_ currently lags behind CO_2_ gas-fed electrolysers in certain performance metrics, it offers a significant improvement in CO_2_ utilization which makes the method worth exploring. This paper presents two simple modifications to a bicarbonate-fed CO_2_ ER system that enhance the selectivity towards CO. Specifically, a modified hydrophilic cathode with Ag catalyst loaded through electrodeposition and the addition of dodecyltrimethylammonium bromide (DTAB), a low-cost surfactant, to the catholyte enabled the system to achieve a FE_CO_ of 85% and 73% at 100 and 200 mA·cm^−2^, respectively. The modifications were tested in 4 h long experiments where DTAB helped maintain FE_CO_ stable even when the pH of the catholyte became more alkaline, and it improved the CO^2^ utilization compared to a system without DTAB.

## 1. Introduction

Carbon capture and utilization (CCU) technologies offer a path for transforming carbon dioxide (CO_2_) into valuable commodity chemicals and fuels, while helping to reduce society’s reliance on fossil fuels [1]. The electrochemical reduction of carbon dioxide (CO_2_ ER) is a technique in which electrical energy (ideally from renewable sources) is used to convert CO_2_ into products such as carbon monoxide (CO), formic acid, methane, etc. [2,3]. Particularly, CO_2_ ER to CO has garnered special interest since it is one of the simplest reaction paths, catalysts such as Ag and Au offer high selectivity, and it is one of the most economically viable products [4,5]. When combined with hydrogen (H_2_), carbon monoxide forms syngas, a versatile product which is the main feedstock for methanol synthesis, or can be transformed to long-chain hydrocarbons via the Fischer–Tropsch process [6,7,8].

As CO_2_ ER to CO has matured, performance metrics have become ever more important for its potential industrial development. We identified five main metrics: (i) current density, (ii) selectivity, (iii) cell voltage, (iv) stability, and (v) CO_2_ utilization. Current density (J) is related to the production rate of CO and H_2_ and a value ≥200 mA·cm^−2^ is recommended for industrial applications [9]. Selectivity, usually evaluated using faradaic efficiency (FE), is the amount of a desired reduction product (in this case CO) compared to the rest of the products, including H_2_ produced from the competing hydrogen evolution reaction (HER). Cell voltage is related to the energy efficiency of the process and a value lower than 3 V is set as a target for the process to be economically competitive (at J > 200 mA·cm^−2^ and FE_CO_ > 90%) [10]. Stability measures the ability of the system to operate at near-constant conditions for a long period of time. Few studies report stability results, with only some reaching the hundred-hour range [11,12,13,14]. Lastly, CO_2_ utilization is the extent to which CO is diluted in unreacted CO_2_ in the gas mixture exiting the reactor (Equation (1)) [15]. This metric is important as it gives an insight into the additional energy needed to separate the CO_2_ ER products from unreacted CO_2_ [16]. This definition is proper if we use bipolar membranes where the crossover is inexistent. That is not the case of anionic membranes that typically present crossover. Unfortunately, this performance metric is underreported or overlooked in most of the CO_2_ ER literature [17].
(1)$${\mathrm{CO}}_{2}\;\mathrm{utilization}=\frac{\left[\mathrm{CO}\right]}{{\left[{\mathrm{CO}}_{2}\right]}_{\mathrm{outlet}}+\left[\mathrm{CO}\right]}\%$$

In recent years, most of the research on CO_2_ ER has focused on using a gaseous stream of CO_2_ as feedstock for the reaction. This method overcomes mass transport limitations and enables systems to achieve higher selectivity towards CO [12,18,19]. High flow rates of CO_2_ are needed to obtain high selectivity, resulting in CO_2_ utilization values below 40% [16], with most of them being around 10% [17,20]. Even in stack systems, the CO_2_ utilization stays below 40% [21]. To tackle this problem, Yuguang Li et al. and Tengfei Li et al. proposed a system in which concentrated aqueous bicarbonate and carbonate were used as the carbon feedstock [22,23]. The CO_2_ is generated in situ at the membrane-cathode interface by the reaction between (bi)carbonate ions and protons produced by a bipolar membrane (Equations (2) and (3)). Then, CO_2_ becomes readily available to be reduced at the cathode’s surface (Equation (4)). This novel method significantly reduces the amount of CO_2_ present in the gaseous output mixture, thereby achieving higher CO_2_ utilization rates [20]. A CO_2_ ER system with (bi)carbonate feed can be coupled more efficiently to an alkaline-based carbon capture system by using the captured solution directly and skipping the energy-intensive regeneration step needed to release the captured CO_2_ [20,24].
(2)$${\mathrm{CO}}_{3}^{2-}+2H^{+}\to {\mathrm{CO}}_{2}+H_{2}O$$
(3)$${\mathrm{HCO}}_{3}^{-}+H^{+}\to {\mathrm{CO}}_{2}+H_{2}O$$
(4)$$\mathrm{CO}2R:{\;\mathrm{CO}}_{2}+H_{2}O+2e^{-}\to \mathrm{CO}+2{\mathrm{OH}}^{-}$$
(5)$$\mathrm{HER}:2H_{2}O+2e^{-}\to H_{2}+2{\mathrm{OH}}^{-}$$

We can highlight many works by D. Sinton et al. [25,26,27] focused on the electrochemical reduction of CO_2_ in the liquid phase from concentrated solutions of bicarbonate and, more recently, carbonate solutions. They normally used bipolar membranes that could generate large amounts of CO_2_ in situ. We can also highlight many works by A. Irabien et al. [28,29,30,31,32,33] centered on the electroreduction of CO_2_ towards high added value products such as methane, methanol, formic acid, ethylene, etc. These works focused on catalysts derived from copper and nickel to enhance the selectivity and efficiency of the CO_2_ ER.

Moreover, we can highlight some literature more focused on the use of silver-based catalysts for the CO_2_ ER. H. Hoshi et al. obtained very interesting results using Ag single crystal for CO_2_ ER to CO [34]. E. Benson et al. studied catalysts derived from rhenium that presented a good selectivity towards CO and partially inhibiting the HER [35]. P. Kang et al. proposed Iridium-derived catalysts for a selective reduction to formic acid [36]. EC. Tornow et al. presented nitrogen and silver organometallic catalysts for a selective reduction to CO [37]. Another interesting example is the recent work by F. Wang et al. [38], where they focused on core–shell metal-based catalysts for electrochemical carbon dioxide reduction, or the work presented by Q. Lu et al. [39] oriented to a nanoporous silver electrocatalyst that was able to electrochemically reduce carbon dioxide to carbon monoxide.

The reduction of CO_2_ to CO occurs at the catalyst surface by the transfer of two protons and two electrons. It has been proven that the most selective catalysts towards the evolution of CO_2_ to CO are gold and silver [3,40], and that is why the literature focuses on these two metals to understand the reduction mechanisms.

The choice of catalyst is one of the most relevant factors to reduce CO_2_ into a given product. It is known that the electrochemical reduction of CO_2_ in the gas and/or liquid phases is a multivariable and complex process. Schematically and according to the bibliography [3,40], the catalysts can be classified according to the final products obtained. This classification schematically summarizes the catalyst selection panel based on the chemical nature of the catalyst, the microscopic structure of the catalyst (nano-structured, etc.), and the support of the catalyst (nanotubes or graphene, for example). In this way, the performance and efficiency of the catalyst will significantly change [41,42,43,44,45].

To obtain “syngas”, silver and gold are the two metals known as excellent catalysts to obtain high selectivity towards CO. Gold is slightly more selective to CO than silver. However, since gold is approximately 50–70 times more expensive than silver [9], many works focused on silver as a catalyst for CO obtention. For organic compounds, copper is one interesting choice to produce hydrocarbons and alcohols at significant current densities, as already indicated in the publications by Hori, Y. et al. [3,40] and Azuma, M. et al. [45], among others. Electrodeposited zinc on copper foil or single atom based on Zn seem to be a good choice for CO_2_ reduction into CO or methane, respectively [46,47].

Finally, the capture of CO_2_ and its storage in geological formations has aroused great interest in recent years, being a very expensive process today and not economically viable [48]. The economic aspect is very important for its implementation in the medium and long term. In this sense, a more interesting option that represents a scientific and technological challenge raises the following question: can we make the transformation of CO_2_ economically interesting? If the answer were yes, a hopeful horizon would open up for our society. CO_2_ by itself has no industrial interest. However, the conversion of CO_2_ to high added value products appears to be a more attractive alternative in comparison to capture and storage. Furthermore, if renewable energies are used in the conversion process, the industrial and economic interest is clear. In this sense, electrochemical reduction is one of the methods that allows CO_2_ to be transformed into useful chemical products with high added value such as carbon monoxide, formic acid, aldehydes, alcohols, and hydrocarbons [2]. The electrical energy used in the reduction process could be derived from different renewable energy sources, such as solar photovoltaic, wind, or hydroelectric sources, which allows this process to have greater flexibility compared to other CO_2_ transformation methods such as the thermochemical [49,50] or the photochemical [51,52]. From an experimental point of view, among the three ways to reduce CO_2_, i.e., thermo-, electro- and photo-catalytic reactions, the electrocatalytic reduction offers conversions higher than 10% [53].

In this paper, we report two simple modifications to the CO_2_ ER system with bicarbonate feed that significantly improved the selectivity towards CO. The first modification proposed was a new procedure which aimed to prepare a Ag-based cathode using electrodeposition on a hydrophilic carbon cloth, which allowed a more effective transport of bicarbonate ions across the cathode towards the membrane. The second modification was the addition of a low-cost surfactant to the catholyte which inhibited the competing HER. These modifications were tested in 4 h long experiments to examine their stability resulting in a useful insight of the system’s steady-state operation.

## 2. Results and Discussion

### 2.1. Electrochemical Cell Assembly and Cathode Electrodes

The experiments were performed in a 4 cm^2^ electrochemical flow cell reactor. The cathode used was either a commonly used hydrophobic gas diffusion electrode (GDE) sprayed-coated with Ag nanoparticles, or a hydrophilic GDE with electrodeposited Ag (Figure 1). The catholyte was an aqueous solution of 2M KHCO_3_ with 20 mM ethylenediaminetetraacetic acid (EDTA) to remove trace metal impurities [54]. For experiments with surfactant, 20 mM of dodecyltrimethylammonium bromide (DTAB) was added to the catholyte. The anode used was nickel foam due to its favorable kinetics for oxygen evolution reaction (OER) and good stability in alkaline media [55]. The anolyte was a solution of 1M KOH. Both catholyte and anolyte flowed through their respective sides at a rate of 10 mL·min^−1^ using two peristaltic pumps. The membrane electrode assembly (MEA) was completed with a bipolar membrane between the anode and the cathode. Under reverse-bias, the bipolar membrane was able to dissociate H_2_O and produce H^+^ and OH^−^ towards the cathode and anode, respectively [56].

The electrochemical cell was operated galvanostatically at current densities of 50, 100, and 200 mA·cm^−2^. The experiments were carried out at 20 °C and at 50 °C as it was proven that higher temperatures enhance CO selectivity [57,58,59]. The gas products that exited the cathodic side of the reactor were accumulated using a phase separator. It is important to note that no gases were present in the catholyte at the entrance of the reactor and all the gases were produced (or released) inside the cell when current was applied. After 300 s of operation, the gas mixture in the phase separator was extracted with a gas-tight syringe and analyzed with a gas chromatography-mass spectrometry (GC-MS) system (Appendix A). For all experiments, the gas mixture was composed only by CO, H_2_, and unreacted CO_2_ (released from the bicarbonate solution). In this work, the selectivity was measured as Faradaic efficiency towards CO (FE_CO_) with the remaining FE going to H_2_ production. A new MEA and fresh electrolyte were used for each experiment.

### 2.2. Cathode of Electrodeposited Ag

CO_2_ ER in gas-fed CO_2_ electrolysers was performed most commonly using a gas diffusion electrode (GDE), which consists of a hydrophobic gas diffusion layer (GDL) and a catalyst layer. The GDL itself was composed of two parts: a macroporous conductive carbon cloth and a carbon microporous layer (MPL). Both components in the GDL were treated with polytetrafluoroethylene (PTFE) to make the materials hydrophobic, preventing the accumulation of water at the cathode (“flooding”) which could lead to a decrease selectivity [60]. Lees et al. found that these two hydrophobic components in the cathode were not suitable for a liquid-fed bicarbonate electrolyser and thus, by removing them, the authors were able to achieve significant improvement in FE_CO_ [15]. The catalyst layer of the GDE is formed by spray-coating the GDL surface with a catalyst ink containing Ag nanoparticles and an ionically conductive ionomer (e.g., Nafion). In another study, Lees et al. found that the content of Nafion ionomer on the catalyst layer was inversely proportionate to the FE_CO_, i.e., the FE_CO_ increased as the GDE Nafion content decreased [61]. The Nafion content can be lowered down to only around 2.6 wt% before the catalyst layer starts to delaminate due to poor adhesion.

Naturally, based on these findings, a cathode that selectively converts CO_2_ to CO in bicarbonate-fed systems should be hydrophilic and should avoid the use of Nafion ionomer. Lees et al. used a hydrophilic GDE without MPL and Ag catalyst loaded through physical vapor deposition (PVD) and spray coated the usual catalyst ink onto it [15]. Zhang et al. used a hydrophilic Ag porous metallic electrode as the cathode [59]. In this work, we present hydrophilic GDE with Ag catalyst loaded through electrodeposition. This method enabled effective CO_2_ conversion to CO in a bicarbonate feed system, while being a facile, relatively cheap, and scalable way of fabricating the electrode.

The electrodeposited (ED) electrodes were fabricated using a GDL containing only the macroporous conductive carbon cloth as the substrate and AgNO_3_ as the silver precursor. The carbon cloth was not treated with PTFE to maintain its hydrophilicity. It should be noted that any gas that was introduced in the reactor was because we generated CO_2_ in situ due to the concentrated bicarbonate solution and the protons provided by the bipolar membrane. Furthermore, it is important to clarify that the catalysts used with Ag NPs were deposited on a cloth containing PTFE (Nafion) and during the preparation of the catalyst, Nafion was also used as a binder. The addition or presence of Nafion in the catalyst is known to add hydrophobicity to the system and would make interaction with bicarbonate ions more difficult. In addition, commercial cloth not treated with PTFE has a hydrophilic character that, predictably, would favor the interaction with bicarbonate ions.

The electrodeposition was carried out by applying −6.3 mA·cm^−2^ to the substrate, and the catalytic loading was adjusted by modifying the electrodeposition time. Two ED electrodes were prepared for the experiments: one with catalytic loading of 1.5 mg·cm^−2^ (ED1.5, Figure 1c,d) and another with 2.5 mg·cm^−2^ (ED2.5, Figure 1e,f). The value of 1.5 mg·cm^−2^ was selected since it was the same loading as the spray-coated (SP) cathode (SP1.5, Figure 1a,b), and 2.5 mg·cm^−2^ to determine whether higher catalytic loading would lead to higher selectivity.

As shown in Figure 1, the ED electrodes formed Ag layers around the carbon cloth fibers. At lower deposition times (i.e., ED1.5), Ag spheres started to grow over the surface of the fiber. Upon longer deposition times (i.e., ED2.5), the spheres grew enough to cover the whole surface, creating a layer of silver upon which a second layer of sphere started growing. In fact, in Figure 1f, it is possible to observe how a portion of the electrodeposited layer was stripped away, leaving the carbon cloth fiber visible. This method enables the fabrication of a pseudo-mesh comparable to the silver porous electrode used by Zhang et al. with a smaller quantity of silver needed [58].

Figure 2 shows that the ED electrodes achieved significantly higher FE_CO_ compared to the commonly used SP electrode. The FE_CO_ increased at all current densities and especially at 20 °C where it more than doubled from 30% for SP1.5 to 70% with ED1.5 at 100 mA·cm^−2^. The increase in selectivity was mainly attributed to an increased production of CO_2_ at the cathode-membrane interface since bicarbonate ions can flow more effectively across the cathode and reach the membrane where they react with H^+^ to produce CO_2_ [15]. This increased CO_2_ production was reflected in the higher [CO_2_]_outlet_ and a higher CO_2_ utilization (Appendix A). An increase in the FE_CO_ with ED electrodes came with a ~400 mV increase in cell voltage compared to the SP electrode at 100 mA·cm^−2^ (Appendix A). This effect could be attributed to a reduction in electrical conductivity in ED electrodes due to the lack of a carbon MPL which was present in the SP cathode.

Most of the research on CO_2_ ER focused on the use of CO_2_ gas as input. This has the advantage of obtaining very high efficiencies and practically inhibiting HER [12,18]. It has the disadvantage of obtaining more than 60% of unreacted CO_2_ as output [11,16,20]. To avoid this, some recent research focused on the use of concentrated solutions of bicarbonate or carbonate without any CO_2_ gas as input. The challenge working in solution was to inhibit HER, which could be the dominant reaction [22,23].

If we compare our obtained results (without adding DTAB) with the existing literature, our FE_CO_ values were close to the published efficiencies. In summary, we obtained around 70% of CO at 100 mA·cm^−2^ which were below the 82% obtained by reference [15], or 95% obtained by reference [59], both working with CO_2_ generated in situ (see Table 1). It is noteworthy that the efficiencies obtained in references [15,59], for example, used Ag catalysts (PVD + NPs) or Ag mesh, which are much more expensive than the Ag ED catalyst proposed in this work. In addition, in some works in the literature, very good efficiencies were obtained but at high pressures (see, for example, reference [59]), which made the experimental set-up more complex and expensive. The values obtained working with CO_2_ gas were higher than 70% and typically higher than 90%, although the operating conditions were completely different from those presented in this work. It should be remembered that the disadvantage of working with CO_2_ gas is that the residual CO_2_ usually exceeds 60%, while we reached values <30% working in solution. Moreover, our results show an efficiency higher than 60% at 200 mA·cm^−2^, an efficiency comparable to the published data working in solution and lower than the results obtained in the gas phase.

### 2.3. DTAB Surfactant in Catholyte

As CO_2_ electrolysers involve two competing reduction reactions, improving the FE_CO_ can be approached from two different strategies: enhancing CO_2_ ER or inhibiting HER. Quan et al. found that several surfactants were able to suppress HER in aqueous electrolytes saturated with CO_2_, and that DTAB achieved the best results [62]. In this study, DTAB was added to the catholyte to study its impact in CO_2_ ER from bicarbonate feed with the three cathodes discussed previously.

Figure 3 and Appendix A show that the addition of DTAB enhanced the FE_CO_ in all the electrodes tested, but the effect was most significant for the SP electrode where it increased by Δ = ~30% at all current densities and temperatures. As for ED cathodes, the increase in FE_CO_ was more modest at 20 °C and improved at 50 °C. The increase in selectivity was attributed to two factors: the first one was the adsorption of DTAB to the cathode’s surface which inhibited HER [62]; the second one was a reduction of the catholyte’s surface tension due to the action of the surfactant and lead to a higher concentration of CO_2_ at the cathode’s interface, reflected in a higher [CO_2_]_outlet_ (Appendix A). A lower surface tension makes it easier for the catholyte to flow through the cathode and reach the membrane, an effect analogous to removing the hydrophobic components of the cathode, which might explain why a bigger effect was observed with the SP electrode. At this point, it is important to highlight that SP cathodes are more hydrophobic than ED ones; for this reason, we hypothesize that the addition of DTAB will improve the hydrophobic cathode (SP) much more than the hydrophilic one (ED).

The cell voltage increased with the addition of DTAB by 50–350 mV depending on the operating current. This was attributed to the inhibition of HER while the CO_2_ ER was left unchanged, which means that higher potentials are needed for CO_2_ ER to replace the current contribution of the inhibited HER and keep the total current constant.

In summary, the addition of DATB decreased the surface tension of the solution, allowing a greater flow of bicarbonate ions to the cathode. This decrease in surface tension lead to a more turbulent flow (visible during the experiments) where the transparent solution became a foam that interacted with the cathode, which will predictably affect the cell potential. The latter, and the decrease in HER, may explain the increase in potential that we observed.

### 2.4. System Stability

A cell with an ED1.5 cathode was tested for 4 h at 50 °C and 100 mA·cm^−2^ to study the evolution in cell voltage, selectivity, and CO_2_ utilization. Figure 4a shows the cell voltage and FE_CO_ of a system without DTAB added to the catholyte. As can be observed, the voltage remained stable around 3.51 V while the FE_CO_ gradually decreased from 70% to 36%. This decline in FE_CO_ is attributed to the increase in catholyte’s pH both at the bulk and at the cathode’s surface [58]. Since CO_2_ ER and HER have OH^−^ as a product (Equations (4) and (5)), the catholyte increased its pH during operation, leading to a shift away from CO_2_ and into a higher concentration of (bi)carbonate in the ${\mathrm{CO}}_{2}/{\;\mathrm{HCO}}_{3}^{-}/{\;\mathrm{CO}}_{3}^{2\;}$ equilibrium.

In fact, the pH of the initial solution was around 8.1 (slightly basic) and during the CO_2_ ER reaction, the CO_2_ generated in situ consumed the bicarbonate ions from the solution by capturing protons from the bipolar membrane and releasing OH^-^ ions to the solution during CO_2_ ER and HER. This gradual basification of the solution shifted the equilibrium towards the region of the carbonate ion and made it more difficult to generate CO_2_ in situ. This increase in pH was related to the decrease in FE_CO_.

At the beginning of the third hour, the catholyte was replaced by a fresh 2M KHCO_3_ solution, the voltage decreased to 3.48 V but rapidly returned to 3.51 V. The FE_CO_ did not return to its initial value, even though the catholyte was replaced to its initial condition which might indicate that the FE_CO_ was determined in greater part by the pH of electrolyte at the cathode’s surface rather than the bulk.

The effect of DTAB on 4 h operation was also studied, obtaining interesting results. As Figure 4c shows, the cell voltage in this system started at 3.59 V and gradually increased to 3.70 V after 3 h of operation. The FE_CO_ began at 79%, dropped to 58% after 40 min and then remained stable around 56% for the rest of the first three hours. Additionally, CO_2_ utilization increased from 26% to 73% after 3 h (Figure 4d), a much bigger increase than the system without DTAB (Figure 4b), indicating a more effective use of the CO_2_. Upon the replacement of the catholyte, the cell potential decreased to 3.66 V but continued to increase from there, while the FE_CO_ improved to 66% and remained stable for the remainder of the time. It appears that since DTAB inhibits HER, the FE_CO_ remains stable even if the catholyte becomes more alkaline, but it comes at a cost of gradually higher cell voltage. The effect was opposite to the system without DTAB, where cell voltage remained stable and FE_CO_ gradually decreased.

It should be noted that the collective losses of cell voltage at 100 mA·cm^−2^ in both systems exceeded the 3V (3.51 V and 3.70 V without DTAB and with DTAB, respectively). The overpotential was still too high to be cost competitive in agreement to reference (3 V at 200 mA·cm^−2^) [10]. The BPM membrane flow cell operates less efficiently in voltage than the AEM membrane. It will be important to identify which cell component (anode, cathode, membrane, or electrolyte) should be optimized to reduce most effectively the overall the cell voltage. Even so, it should be noted that both FE_CO_ and %CO_2_ utilization with DTAB showed very relevant results.

Finally, an intermittency test was carried out to discard catalyst deactivation as the cause of decreasing FE_CO_. The test consisted in stopping the current for 5 min and then reapplying 100 mA·cm^−2^. In both systems, the cell potentials and FE_CO_ reached values close to their initial values at the start of the 4 h experiment, and in the case of the system with DTAB, the FE_CO_ reached 88%, which was higher than the initial FE_CO_ recorded. These results indicate that the Ag cathode did not suffer deactivation. Instead, we believe that the decrease in FE_CO_ was caused by a pH gradient formed during the steady-state operation, which lead to a much more alkaline pH at the cathode’s surface compared to the catholyte’s bulk pH. These results highlight the importance of stability tests since they can help identify not only deficiencies in a catalyst’s durability but also system-related limitations that affect the selectivity and efficiency of the process in the long term.

In summary, if we compare our results obtained (with DTAB) with the existing literature, our FE_CO_ values were among the best efficiencies obtained, around 85% at 100 mA·cm^−2^ and higher than 70% at 200 mA·cm^−2^ (see Table 1). These excellent efficiencies were obtained under reasonably simple operating conditions, working at atmospheric pressure, in conditions of 50 °C, and with cathodic catalysts based on low-cost electrodeposition techniques.

## 3. Materials and Methods

KHCO_3_ (99%) and ethylenediaminetetraacetic acid (EDTA) were purchased from Scharlab (Barcelone, Spain). KOH (85%) was purchased from Labbox (Barcelone, Spain). Nickel foam (99.99%) was purchased from Nanografi Nano Technology. Fumasep FBM bipolar membranes were purchased from FuMa-Tech (Bietigheim-Bissingen, Germany). Silver nanoparticles (<100 nm, 99.5% trace metals basis), Nafion solution (5wt%), and AgNO_3_ (≥99.0%) were purchased from Sigma-Aldrich. GDL carbon cloth with carbon MPL and treated with PTFE was purchased from Fuel Cell Store (Bryan, TX USA). GDL carbon cloth without MPL and untreated was purchased from Quintech (Göppingen, Germany). Dodecyltrimethylam-monium bromide (DTAB) was purchased from Glentham Life Sciences (Corsham, United Kingdom).

The electrochemical flow cell was purchased from ElectroChem Inc. (Raynham, MA, USA). The electrochemical flow cell contained two graphite flow plates pressed together by two current collector plates with a gold coating, which act as housing of the reactor. Silicon and PTFE gaskets were pressed between the flow plates for water- and gas- tightness. The temperature was modified through resistive heating attached to the housing using temperature controller Eurotherm 2408. The electrolytes flowed using two peristaltic pumps Dinko Instruments D-25V. The electrochemical measurements were carried out using a Autolab PGSTAT302N potensiostat/galvanostat. The gases were analyzed using a GC (Varian 3900 with Carboxen-1006 PLOT Column) attached to an MS (Pfeiffer Vacuum Hi-Cube) using Argon as the carrier gas. The gas samples from the system were extracted using a 5 mL SGE gas-tight syringe.

Spray-coated (SP) electrode preparation. The catalyst ink was as proposed by Verma et al. [18], prepared by mixing 42 mg of Ag nanoparticles (<100 nm, 99.5% trace metals basis, Sigma Aldrich (Darmstadt, Germany)), 55 μL Nafion 5% solution (5 wt%, Sigma Aldrich (Darmstadt, Germany)), 1600 μL of deionized water, and 1600 μL of isopropyl alcohol (2-Propanol, LabKem). The ink was sonicated (Selecta Ultrasons (Santiago de Compostela, Spain)) for 20 min. The ink was spray-coated onto a hydrophobic carbon cloth GDL with carbon MPL (Fuel Cell Store (Bryan, TX USA)) with an area of 4 cm^2^ (2 × 2 cm^2^) until it reached a catalyst loading of 1.5 ± 0.2 mg·cm^−2^. The catalyst loading was determined by weighing the GDL before and after deposition.

Electrodeposited (ED) electrode preparation. The electrolyte for electrodeposition was a solution of 1 M KHCO_3_ (to increase conductivity), 0.01 M of EDTA to remove impurities, and 0.05 AgNO_3_ as Ag precursor. The electrodeposition was performed in a custom-made electrochemical cell where the hydrophilic carbon cloth GDL (without MPL) was pressed against a stainless-steel plate (that served as current collector) and introduced into the electrolyte. The carbon cloth worked as cathode and a graphite carbon rod as anode. The electrodeposition was performed with a chronopotentiometry at −6.3 mA·cm^−2^ for 380 s for ED1.5 and 600 s for ED2.5. The electrodeposited cathode was thoroughly rinsed after deposition to remove the electrolyte from the cloth and then dried at 60 °C using a plate heater. The catalyst loading was determined by weighing the (dry) carbon cloth GDL before and after the electrodeposition.

## 4. Conclusions

In this paper, two proposed modifications to a CO_2_ ER bicarbonate-fed electrolyser system were studied. Namely, a silver-based electrodeposited cathode and DTAB surfactant additive to the catholyte were able to enhance the FE_CO_ of the process. Both modifications boosted the FE_CO_, reaching a value of 73% at 100 mA·cm^−2^ and 47% at 200 mA·cm^−2^ at room temperature, and 85% at 100 mA·cm^−2^ and 73% at 200 mA·cm^−2^ at 50 °C. These results are among the highest selectivities reported in literature for CO_2_ ER systems with (bi)carbonate feed and competitive with gas-fed CO_2_ electrolysers (Table 1). We present these modifications as tools that can be applied independently to this type of electrolyser to improve its performance and move a step closer towards industrially relevant operating conditions. Despite the two improvements that we proposed in this work, there are still important issues to solve, which we presented in the stability tests section: the cell potential obtained with the bipolar membrane remains high for scale-up application; the cell potential increases over the time; and the basicity of the solution increases over time, reducing the amount of CO_2_ produced in situ.

## Figures and Tables

**Figure 1 molecules-28-01951-f001:**
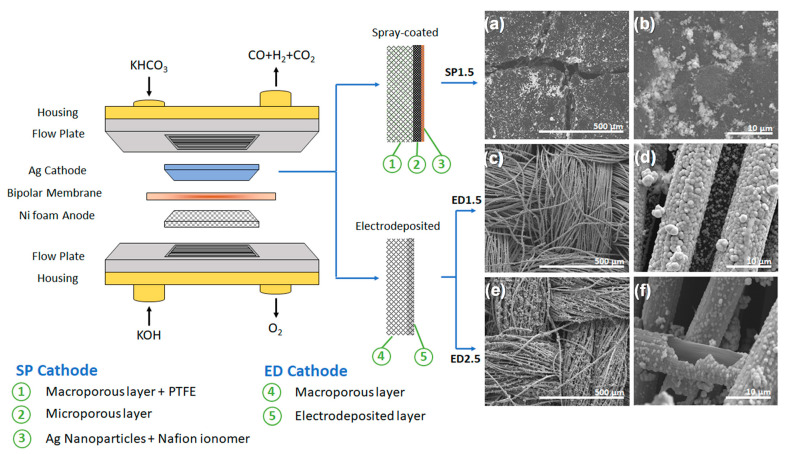
Electrochemical cell assembly and its components. Three different Ag cathodes were tested: one containing spray-coated Ag nanoparticles (SP1.5) and two with electrodeposited Ag (ED1.5 and ED2.5). The cathodes were observed through SEM: (**a**,**b**) SP1.5, (**c**,**d**) ED1.5, (**e**,**f**) ED2.5.

**Figure 2 molecules-28-01951-f002:**
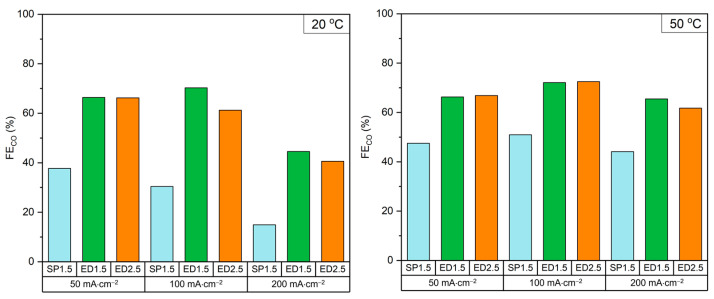
FE_CO_ of the three Ag cathodes tested at 20 °C and 50 °C.

**Figure 3 molecules-28-01951-f003:**
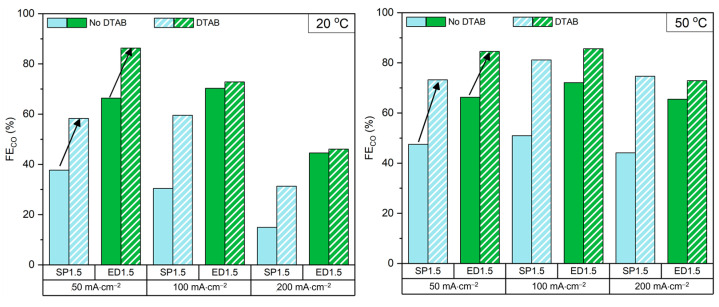
FE_CO_ of the SP1.5 and ED1.5 cathode, with and without DTAB in the catholyte, at 20 °C and 50 °C.

**Figure 4 molecules-28-01951-f004:**
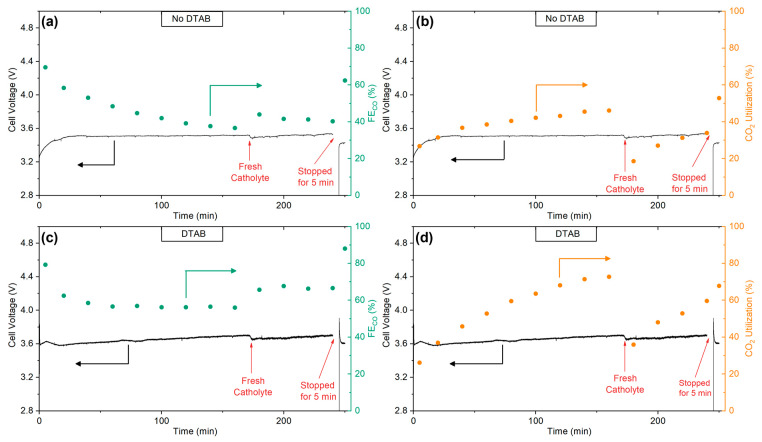
Cell potential, FE_CO,_ and CO_2_ utilization at 100 mA·cm^−2^ of a system with ED1.5 as cathode, 50 °C and FBM bipolar membrane (**a**,**b**) without DTAB and (**c**,**d**) with DTAB added to the catholyte.

**Table 1 molecules-28-01951-t001:** Faradaic efficiency towards CO of CO_2_ ER systems reported in the literature and their operating conditions.

CO_2_ Feedstock	Catalyst	Membrane	J (mA·cm^−2^)	FE_CO_	Temp	Pressure	Reference
2M KHCO_3_ with 0.02M DTAB	Ag ED	BPM	100	70%	20 °C	1 atm	This work
			200	45%	20 °C	1 atm	
			100	85%	50 °C	1 atm	
			200	73%	50 °C	1 atm	
2M KHCO_3_	Ag NP	BPM	100	40%	50 °C	1 atm	[57]
			200	46%	50 °C	1 atm	
3M KHCO_3_	Ag XXXXXX(PVD + NP)	BPM	100	82%	RT	1 atm	[15]
			200	62%	RT	1 atm	
3M KHCO_3_	Ag foam	BPM	100	59%	20 °C	1 atm	[59]
			200	~34% *	20 °C	1 atm	
			100	95%	20 °C	4 atm	
			100	78%	70 °C	1 atm	
1M K_2_CO_3_	Ag NP	BPM	100	28%			[22]
			200	~20% *			
3M KHCO_3_	Ag NP	BPM	100	37%			[23]
CO_2(g)_	Ag NP	BPM	100	67%			[55]
			200	50%			
CO_2(g)_	CoPc	AEM	200	88%			[63]
CO_2(g)_	Ag NP	AEM	200	>90%	RT		[64]
CO_2(g)_	Ag NP	-	417	100%			[16]

* Obtained from graphical data. NP = nanoparticles. RT = room temperature. BPM = bipolar membrane. AEM = anion exchange membrane. ED = electrodeposited. PVD = physical vapor deposition. ~ = approximately.

## Data Availability

Not applicable.

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
