# Peer review of "Strategies to Enhance CO2 Electrochemical Reduction from Reactive Carbon Solutions"

_molecules, 2023, doi:10.3390/molecules28041951_

Round 1
Reviewer 1 Report
The article describes the current state of CO2 utilization research. Manuscript is well-structured and in good shape. However, it is advised to make minor changes before publication in "Molecules." Here are my findings.
Comments:
1. BE consistent when using acronyms such as FECO or FECO.
2. Suggest utilizing CO2ER instead of CO2R to clearly identify between CO2 electrochemical reduction and other reduction types.
3. Recommended to include literature on alternative approaches for CO2 utilization in order to highlight the significance of Electrochemical CO2R and to delve into challenges linked to these techniques in order to make the problem statement clearer.
4. Introduction lacks to provide a convincing argument for the use of Ag as a CO2R catalyst in comparison to other metals like Cu, Au, Zn, etc. Include a comparison analysis to back up Ag's role as a powerful CO2R catalyst.
5. Please eliminate/reduce the use of redundant words. E.g., also in order, however etc. Please edit all instances where these terms appear, as deleting them would not significantly alter the sentence's meaning.
6. Overpotential during the reaction has a detrimental effect on CO2R. No such information is supplied in the text. However, desirable criteria include a low overpotential and a high FE. Advised to emphasize.
7. Chronopotentiometry was conducted at a negative (-6,3 mAcm-2) current density, while the exact opposite current density was applied for Ag electrodeposition. What is the rationale for choosing a given current density value in both circumstances, and how is it determined/fixed? Are alternative current densities applied to observe the influence on linked parameters?
8. Fig. resolution need to improve i.e., Fig 3, 4.
9. Suggested to include measured size of Ag nanoparticles using TEM since particle size is quite significant in terms of the number of active sites directly affecting the rate of surface-catalyzed reactions.
10. Line260-265, “ gradual basification of the solution shifts the equilibrium ……..” however no experimental data available for CO2R at conc. lower than 1 M HCO3.
Reviewer 2 Report
The authors evaluated the performances of two simple modifications for CO2 electrochemical reduction system. The manuscript was well written and the result is good. I recommend it for publication if the authors objectively evaluated the shortcomings of the modifications in Conclusion.
